# Co-developing a comprehensive disease policy model with stakeholders: The case of malaria during pregnancy

**Silke Fernandes**[1]*, **Andrew Briggs**[2], **Kara Hanson**[1]

**1** Department of Health Services Research and Policy, Faculty of Public Health and Policy, London School of Hygiene and Tropical Medicine, London, United Kingdom, **2** Department of Global Health and Development, Faculty of Public Health and Policy, London School of Hygiene and Tropical Medicine, London, United Kingdom

* silke.fernandes@lshtm.ac.uk

## Abstract

Understanding the holistic impact of malaria during pregnancy is essential for improving maternal and child outcomes in malaria endemic settings. To be able to design appropriate research and conduct robust policy analyses, a comprehensive model of the underlying disease, representing the current understanding of mechanisms and consequences, is needed. This study aimed to illustrate a methodology to co-develop a disease policy model with expert stakeholders using malaria during pregnancy as a case study. An initial steering group was convened to develop a first model of malaria during pregnancy and its consequences for mother and child based on their understanding of the literature. Subsequently, this model was refined using a Delphi process to gain consensus amongst twelve experts working in the field of malaria during pregnancy, representing the disciplines of health economics, mathematical modelling, epidemiology and clinical medicine. The experts reviewed drafts of the conceptual model and provided feedback in two rounds of semi-structured questionnaires with the aim of identifying the most important health outcomes and relationships in both mother and child as well as the most relevant stratifiers for the model. Final consensus on any areas of disagreement was reached after two online meetings. The final model is a comprehensive disease policy model of malaria during pregnancy, including ten maternal and ten child health outcomes with four stratifiers. The model developed in this study should be of value to malaria researchers, funders, evaluators and decision makers, though some adaptation will be required for each specific context and purpose. In addition, the methodology and process followed in this study is replicable and can guide researchers aiming to develop a conceptual model for other conditions. The model resulting from this study highlights the complexity required to depict fully the consequences of malaria during pregnancy for both the mother and the child. It also demonstrates how to conduct a rigorous process to develop a

**Data availability statement:** All relevant data are within the paper and its Supporting Information files.

**Funding:** This work forms part of the PhD of SF supervised by KH and AB and the IMPROVE and IMPROVE-2 studies, through which SF was funded to conduct the cost-effectiveness component. The IMPROVE and IMPROVE-2 studies received financial support from the EDCTP2 programme under Horizon 2020 (TRIA.2015-1076, TRIA.2015-1076b); the UK Department of Health and Social Care, the UK Foreign Commonwealth and Development Office, the UK Medical Research Council, and Wellcome Trust, through the Joint Global Health Trials scheme (MR/P006922/1); and the Swedish International Development Cooperation Agency. The funders had no role in study design, data collection and analysis, decision to publish, or preparation of the manuscript.

**Competing interests:** The authors have declared that no competing interests exist.

disease policy model. In addition, the study has helped to identify a number of areas with scarce data and need for further research.

## Introduction

Most diseases work via complex biological processes that create observable inter-related health outcomes. Understanding these relationships is essential for many types of research and the embedded policy analyses that are informed by that research. For example, trials of comparative effectiveness and associated cost-effectiveness analysis would benefit from an understanding of all the relevant outcomes, and the interconnections among them, both to design the most appropriate research study and to analyse the results of that study to understand which treatment options are most appropriate in a given context.

Global and national bodies often require a cost-effectiveness analysis before adopting a new intervention. Such analysis allows a comparison between alternative uses of scare resources.

Because health interventions can have multiple health effects (for example, both morbidity and mortality), composite metrics such as Quality Adjusted Life Years (QALYs) or Disability Adjusted Life Years (DALYs) are frequently used. Complexity is increased when a disease affects more than one population group.

Malaria during pregnancy has been shown to be associated with a wide range of outcomes affecting both, the mother and the child. Outcomes in the pregnant woman include clinical malaria, anaemia and increased maternal mortality. In the child, malaria during pregnancy can lead to, amongst other outcomes, stillbirth, low birth weight and increased <5 mortality [1]. In this paper, we use malaria in pregnancy as a case study to illustrate how a structured approach to co-developing a disease policy model with relevant stakeholders can result in a more robust model, which will carry greater influence with the scientific and policy community because of the multi-disciplinary input into its development.

In 2012 a taskforce recommended the development of such conceptual models, which we term disease policy models, as the foundation for developing an economic model [2]. A disease policy model entails a systematic approach to developing a visual framework for analysis that shows how specific health outcomes and pathways relate and interact with each other [2]. It is different from disease transmission models, which use mathematical equations to predict or explain the processes underlying disease transmission. It is also distinct from most cost-effectiveness models which do not necessarily follow a systematic procedure to design the model used to estimate both costs and benefits associated with two or more interventions. However, the robustness and generalisability of cost-effectiveness models is enhanced by being based on a disease policy model. Documented approaches to the development of conceptual frameworks include literature reviews, consultation with stakeholders (qualitative and quantitative), methods of incorporating stakeholder views and piloting to refine the framework [3,4].

Economic evaluations of interventions can be complex and require contributions from a broad range of disciplines. Models based on a particular viewpoint can lead to poor validity and credibility. A review of outcomes included in published economic models of malaria in pregnancy interventions found that studies used a wide range of different outcomes in estimating the DALYs averted [5–13]. Only four out of nine CEAs incorporated clinical malaria, maternal anaemia and low birth weight [6,7,10,12], which are commonly measured in clinical trials, in the DALY estimation. One CEA did not include any child health outcome [9] and another no maternal health outcome [5].

One approach for integrating data from a range of perspectives is the Delphi consultation method. This is a well-established and tested approach used in research to elicit information from experts and has been used extensively in the social sciences [14–17].

It is particularly suitable when it is necessary to incorporate a range of stakeholder views, which in turn can lead to improved quality and acceptability of an economic evaluation model and its findings[2,18–20]. Such a process of generating consensus amongst experts avoids the pitfall of only including the outcomes and relationships measured in trials.

The aim of this study was to co-develop a disease policy model of *P. falciparum* malaria during pregnancy for pregnant women and their babies using a Delphi consensus study with experts in the field of malaria during pregnancy. The expert panel's task was to identify the most important health outcomes and relationships in both mother and child and the most relevant stratifiers for the model. In doing so, the study demonstrates that co-production of holistic disease models with expert stakeholders, representing the current understanding of a disease and potential treatment pathways, is feasible and represents a more robust approach than *ad hoc* model construction by individual academic teams.

## Methods

This study used the Delphi methodology to co-develop a disease policy model of malaria during pregnancy with expert stakeholders. The expert panel in the Delphi methodology consists of people with relevant insight into the subject and can include technical experts, health providers, policy makers, patients or other suitable panellists. It is a particularly useful technique to gather input from various stakeholders in a time-efficient manner through a series of questionnaires. Responses from each round are collated, analysed and incorporated into the subsequent rounds of questions until consensus between the panellists has been reached, usually after two to three rounds, which is often followed by a final consensus meeting with stakeholders to resolve any final disagreements [16,17,21]. The experts remain blinded to each other's identity in the process up until the final meeting (if applicable), which promotes equal contribution independent of status and other factors and removes less favourable forces of group dynamics [16,17].

The different stages and methods in this study are illustrated in Fig 1 and summarized below. Full details of the approach are described in the supplementary materials. Experts in this study were first approached on 31 August 2022 and twelve experts consented by 5th of October 2022. The final consensus meeting took place on 8th September 2023.

### Stage one: Preparation

During an initial preparation stage, a steering group with collective experience in health economics, disease policy modelling and epidemiology of malaria in pregnancy was convened. Its task was to short-list experts to be approached to be part of a Delphi panel as well as to advise on the preparation of a first draft of the disease policy model and questionnaire based on their understanding of the literature, ongoing research and natural history of malaria in pregnancy. Potential candidates for the Delphi panel were purposively sampled for their varied expertise, knowledge of the literature and current research in malaria during pregnancy and approached by email. The authors paid particular attention to having a well-balanced panel with experts representing both maternal and child health, early and later exposure to malaria during pregnancy and various endemicity contexts. The study team aimed to include eight to ten experts in the Delphi panel, a group size shown to be effective and reliable for the Delphi method [16,17]. Experts received no incentive or financial reimbursement for their time participating in this study.

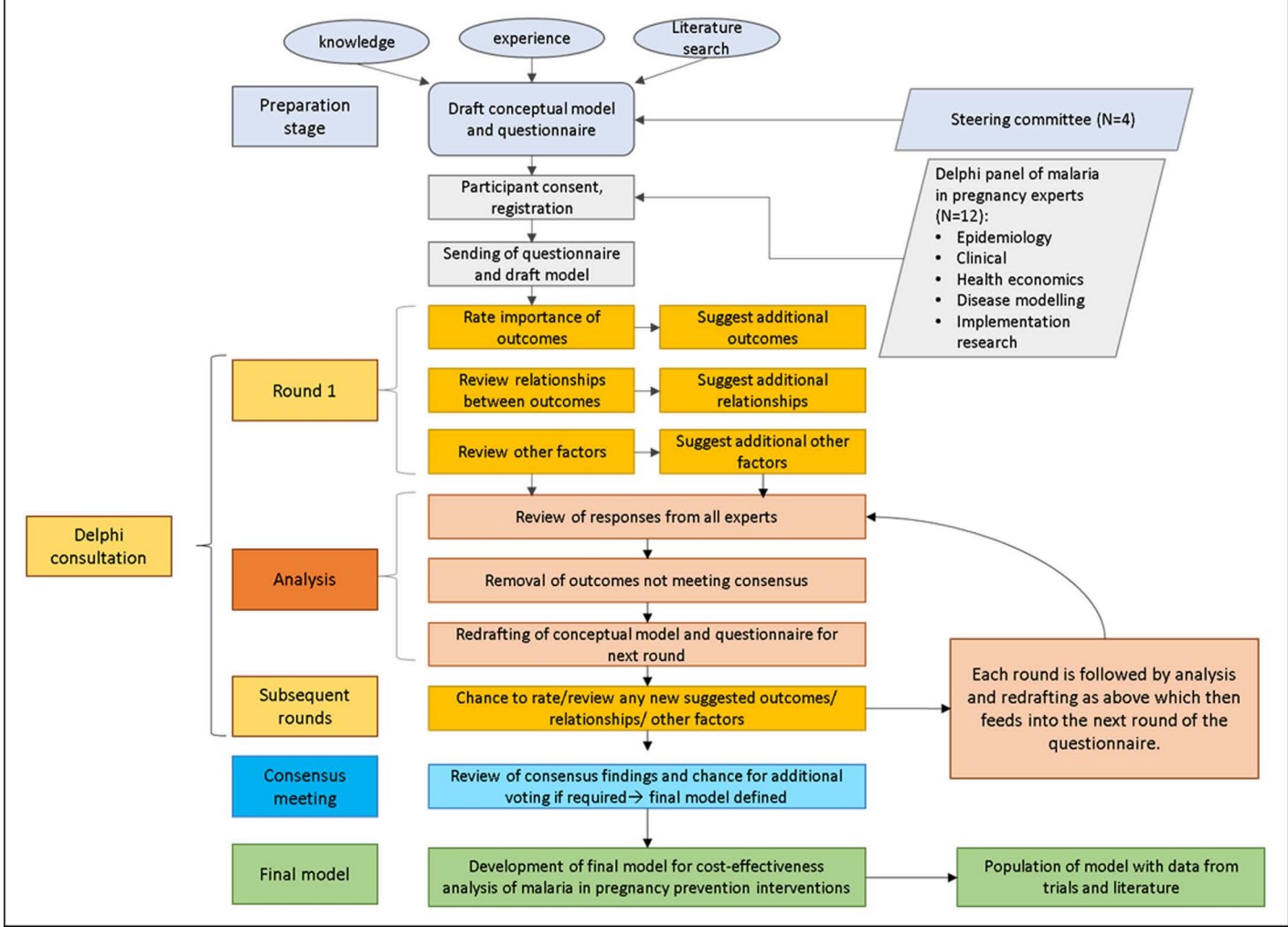

**Fig 1. Methodology used in the study.** Fig 1 illustrates the methodology used in this Delphi consultation study, which can be divided into three stages: preparation, Delphi consultation and consensus meeting.

## Stage two: Delphi consultation

Twelve experts agreed to take part in the Delphi study and provided written informed consent (online). During the Delphi consultation stage, they were asked to refine the draft model using an *a priori* undetermined number of rounds of consultations until consensus in most questions was reached. The threshold for consensus for individual questions was set at 70% of panellists agreeing, consistent with previous Delphi studies [19,22]. In each round, the study team provided the panel members with a current draft of the model and asked them to perform the following tasks:

1) assess importance of the outcomes in the model,

2) suggest additional outcomes that were missing,

3) evaluate the accuracy of relationships between outcomes,

4) suggest additional relationships between outcomes that were missing,

5) order stratifiers for subgroup analysis by importance (a stratifier is defined as a variable which can partition the population in the model into subpopulations, e.g., by gravidity or HIV status of the mother),

6) suggest any additional stratifiers that were missing, and

7) provide their opinion on additional aspects of the presentation of the model, e.g., the visual presentation of low birth weight with its sub-categories (prematurity, intrauterine growth restriction and small for gestational age).

Nominal and ordinal categorical response options as well as free-text questions were used in both questionnaires with round two containing more of the latter (See S2 Text and S3 Text for questionnaires used in round 1 and 2). The responses to each round were analysed by one researcher (SF) and incorporated into the next model draft and questionnaire. Categorical questions were analysed using simple descriptive statistics. Free text responses were explored using a simple thematic analysis, coding them manually into themes [23]. After each round panellists received a summary report of the analysis (See S4 Text and S5 Text), ensuring anonymity was maintained.

### Stage three: Consensus meeting

In the final stage of the study, two online consensus meetings for experts in different time zones were held to present the findings of the second Delphi round and to discuss and vote on any remaining aspects of the model where consensus had not been reached during stage two. The disease policy model was finalized by the first author (SF) following the consensus meetings.

### Ethics

Ethics approval for this study was received on 12 July 2022 by the Research Ethics Committee of the London School of Hygiene and Tropical Medicine (Reference number 27361). Informed written consent was received from all Delphi panel members.

## Results

Key results of the different stages of this study are summarized here (see supplementary materials for additional detail).

### Stage one: Preparation

The first draft of the disease policy model developed by the steering group consisted of outcomes for the mother, the child and the relationships between included outcomes (Figs 2 and 3). Gravidity, timing of exposure to *P.falciparum* (i.e., first, second or third trimester) and HIV status were selected as the most relevant stratifiers for subpopulation analysis. The steering group identified 17 experts to be approached to participate in the study, of whom twelve agreed (71%). Amongst eleven experts the average years of experience working in malaria in pregnancy was 17.9 years (range 8–34) and the twelfth expert had over 15 years of experience in the economics of malaria.

### Stage two: Delphi consultation

Two rounds of consultation were required before sufficient consensus was reached. Changes to the model made after each consultation round and the consultation meetings are illustrated in table 1 with only the most significant changes highlighted here in the text.

After the first consultation round all outcomes included in Figs 2 and 3 remained in the model. On recommendation of panel members "severe disease" and "serious complications" were combined into the single outcome "severe malaria" as experts pointed out the overlap between the two and hence the difficulty in differentiating between these two outcomes. All experts agreed that "low birth weight" should be separated into "intrauterine growth restriction" and

"preterm birth", with five experts suggesting the addition of "small for gestational age". Additional maternal) to be incorporated into the next draft of the model were "asymptomatic parasitaemia", "placental malaria" and "hypertension disorders of pregnancy". Responses on rating stratifiers (gravidity, HIV status, timing of exposure) were inconclusive and needed further exploration. Seven experts suggested that "transmission intensity" be added as an additional stratifier.

In round two experts reached consensus regarding the inclusion of asymptomatic parasitaemia (100%, 12/12 agreed) and placental malaria (75%, 9/12 agreed) and their associated relationships. Whether to include or exclude "hypertension disorder of pregnancy" was unclear and had to be scrutinized further during the consensus meetings. While all relationships associated with "hypertension disorder of pregnancy" were judged to be correct, it appeared there was a difference in opinion regarding its importance and relevance amongst experts.

Experts were asked to vote for the two most important stratifiers for subpopulation analysis leading to the following ranking from most to least important with the number of votes in brackets (one expert only voted once): gravidity (10), transmission intensity (8), timing of exposure of P.falciparum (3) and HIV status (2). Summary reports of both Delphi consultation round analyses can be found in S4 Text (round 1) and S5 Text (round 2) and intermediate model drafts after rounds 1 and 2 are depicted in S1 Text (Figs A and B).

### Stage three: Consensus meeting

All twelve experts completed both rounds of questionnaires and nine (75%) attended one of the consensus meetings, held on 31st August and 8th of September 2023. The most relevant topic discussed was "hypertension disorders of pregnancy" and its potential sequelae. All attending experts agreed to keep "hypertension disorder of pregnancy" in the model without splitting it further into "hypertension", "pre-eclampsia" and "eclampsia". However, they voted to add "long-term effects of hypertension disorders of pregnancy" as a further outcome to include long-term sequelae such as stroke or mental health disorders. Other, less contentious issues such as the relationship between "clinical malaria" and "anaemia" or relationships and labelling of child morbidities were also agreed during the consensus meeting.

Experts expressed the importance of adapting economic models to context and allowing flexibility for them to evolve over time as more granular data become available. They also felt that in addition to developing a disease policy model of malaria during pregnancy to be used in future cost-effectiveness analyses, the work had helped to identify a number of areas where data are limited which will be important to share with the research community (A summary of the consensus meetings can be found in S6 Text). The final model is shown in Fig 4, in which both child and maternal figures are combined, a suggestion made during the consensus meeting.

## Discussion

### Summary

This article presents a consensus-building study using the Delphi methodology with the goal of co-developing with expert stakeholders a disease policy model of malaria during pregnancy that can be used to inform trial design and analyses to inform malaria policy development. The result is a comprehensive disease policy model that includes ten maternal and ten child health outcomes, as well as four stratifiers. To our knowledge, it is the first formal attempt to co-develop a disease model of this kind either in the field of malaria or in a disease area predominantly prevalent in low- and middle-income countries.

The study has highlighted the complexity of the model required to depict the full set of consequences of malaria during pregnancy for mothers and their offspring. Key contributors to the success of the study were the selection of the expert panel, thorough preparation of each stage, and careful analysis and weighing up of all responses. It was essential to be accurate with language, which sometimes had to evolve over various stages, while remaining accessible to a wide range of readers.

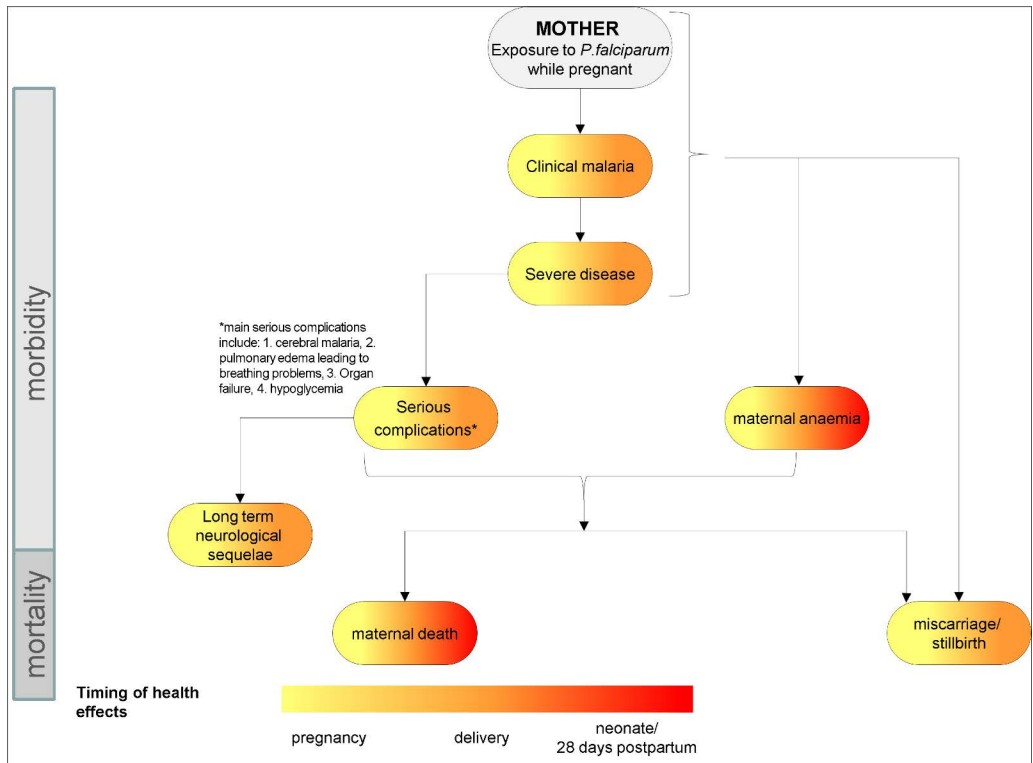

**Fig 2. Disease policy model maternal outcomes: Draft 1 – steering group.** Fig 2 show the maternal (Fig 2) included in the first draft of the disease policy model developed by the four members of the steering group, which was used as the starting point for the first round of the Delphi consultation. Outcomes were divided into maternal (Fig 2) and child (Fig 3) outcomes. Outcomes affecting morbidity are shown at the top, while mortality outcomes at the bottom. The colour coding of the outcomes represents the different timings of the health effects.

The process not only helped to develop the model to include relevant outcomes and relationships but also improved the visual presentation and accessibility of the model, for example by adding symbols for the different timings of outcomes or appearance of arrows.

## Strength and limitations

This study has a number of strengths and limitations. The literature search conducted by the first author during the preparation stage was not a systematic review. Therefore, some potential outcomes and relationships may have been missed out of the first draft of the model, however this was mitigated by the experts' responses during the consultation rounds and consensus meetings. Experts for the Delphi panel were purposively selected to balance the experience, origin and focus area of work of panel members, however, the study may suffer from bias by omitting other experts with differing views.

The acceptance rate of experts was high (71%) with a 100% retention during the two consultation rounds; and 75% of panellists attended one of the two consensus meetings at the end of the process. The use of the Delphi methodology preserved the anonymity of experts and allowed panellists to respond freely without being influenced by other opinions or dominant personalities. The final stage of the study using online consultation meetings was more susceptible to the effects of group dynamics, however, this did not appear to be a problem with all experts engaging equally and respectfully with each other in both meetings.

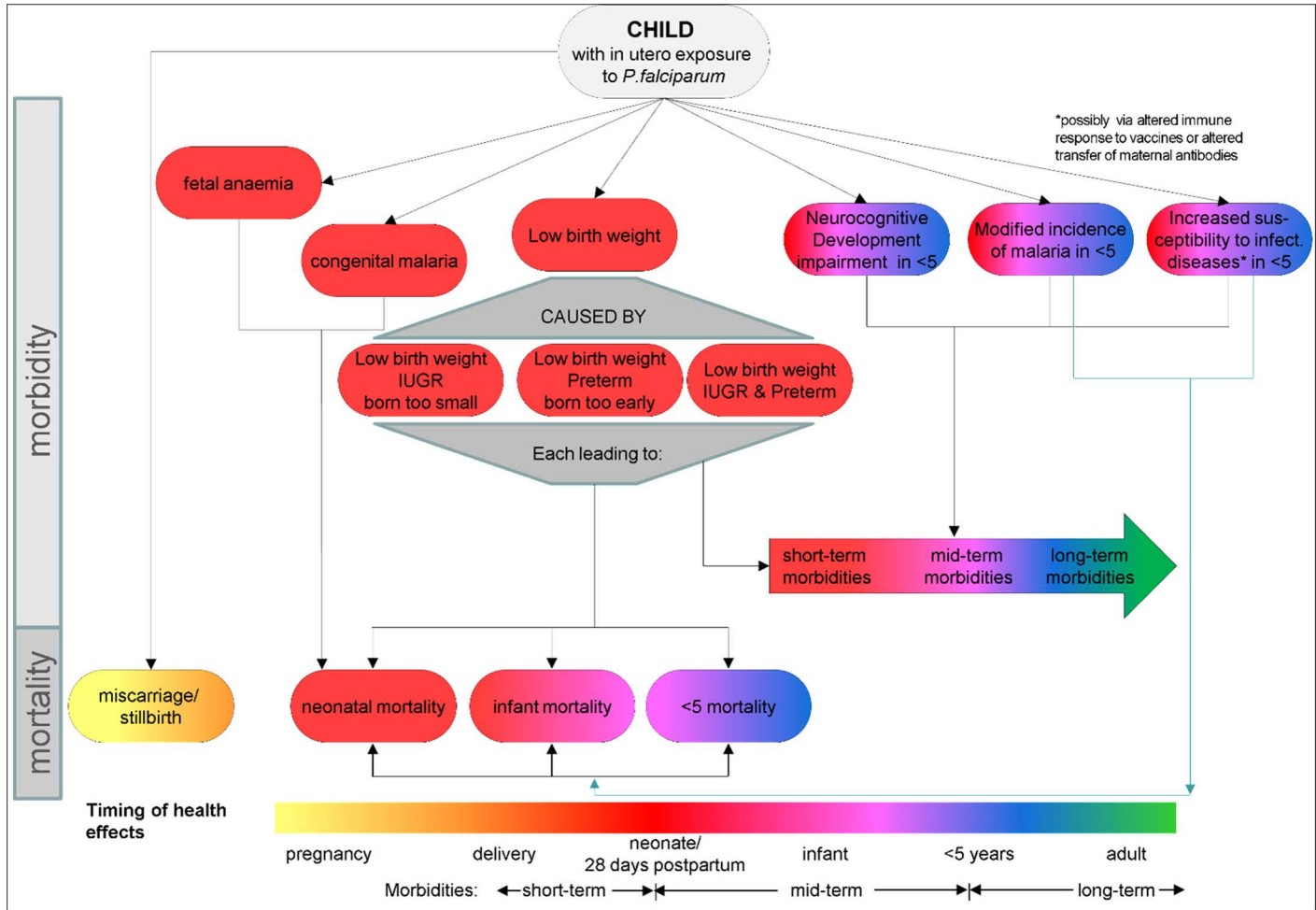

**Fig 3. Disease policy model child outcomes: Draft 1 – steering group.** Fig 3 show child outcomes (Fig 3) included in the first draft of the disease policy model developed by the four members of the steering group, which was used as the starting point for the first round of the Delphi consultation. Outcomes were divided into maternal (Fig 2) and child (Fig 3) outcomes. Outcomes affecting morbidity are shown at the top, while mortality outcomes at the bottom. The colour coding of the outcomes represents the different timings of the health effects. Abbreviations: IUGR = Intrauterine growth restriction.

This study focused on *P. falciparum* malaria. To apply the model to other plasmodium species such as *P. vivax*, *malariae* or *ovale* it would be necessary to review the model and consider inclusion of additional outcomes, relationships and stratifiers, informed by a literature search and expert consultation.

It may be a challenge to populate this comprehensive model for a cost-effectiveness study because of the range of outcomes and complexity of relationships. Nevertheless, this study has brought together experts from different fields and contexts to develop a model all could agree to.

## Areas for future research

During the study, a number of areas requiring further research or development emerged. The most commonly used outcome in cost-effectiveness analysis of global health interventions is the DALY. It is a composite outcome combining mortality and morbidity, and in the case of malaria during pregnancy can combine both maternal and child outcomes into one measure. However, not all outcomes lend themselves equally well to calculating reliable DALY estimates and all of them

**Table 1. Changes to the model made after each consultation round and consultation meetings.**

| | | Changes made after Delphi consultation round 1 | Changes made after Delphi consultation round 2 | Changes made after Delphi consultation meetings (final model) |
|---|---|---|---|---|
| Mother | Outcomes | "MOTHER Exposure to P.falciparum while pregnant" changed to "MOTHER presence of P.falciparum while pregnant" | | |
| | | Addition of "asymptomatic parasitaemia" | | |
| | | "With and without Placental malaria" added to "clinical malaria" and "asymptomatic parasitaemia" | | |
| | | "Long-term neurological sequelae" changed to "long-term sequelae" to include other long-term disabilities resulting from manifestations of severe malaria | "Long-term sequelae" changed to long-term neurological and other sequelae" | green colour (for long-term consequence to the mother) added to "long-term neurological and other sequelae" |
| | | "Death in utero after maternal death" added to "miscarriage/stillbirth" | "miscarriage/stillbirth/death in utero after maternal death" changed to "miscarriage/stillbirth/death in utero" | |
| | | "Hypertensions disorders of pregnancy" added | Outcome "hypertension disorders of pregnancy" relabelled to "hypertension disorders of pregnancy and post-partum", footnote added: "includes pre-eclampsia, eclampsia and gestational hypertension" | Additional outcome "Long-term effects of hypertension disorders of pregnancy" added as a consequence of "Hypertension disorders or pregnancy and post-partum" |
| | | "Severe disease" and "serious complications" combined into "severe malaria" and WHO definition added | | |
| | Relationships | Relationships to and from "hypertension disorders of pregnancy" added | | Relationship from "Hypertension disorders of pregnancy and post-partum" to "Long-term effects of hypertension disorders of pregnancy" added |
| | | Relationship from maternal death to "death in utero after maternal death" added | | |
| | | Relationship to and from "asymptomatic parasitaemia" added | | |
| | | Relationship from maternal anaemia to "severe malaria" added | | |
| | | Relationship between maternal anaemia and clinical malaria made bi-directional | Bidirectional relationship between "clinical malaria" and maternal anaemia" reversed to unidirectional (malaria to anaemia) and an arrow from "maternal anaemia" to "clinical malaria" indicating "contributes to progression" | Arrow from "maternal anaemia" to "clinical malaria" indicating "contributes to progression" removed |

*(Continued)*

|  |  | Changes made after Delphi consultation round 1 | Changes made after Delphi consultation round 2 | Changes made after Delphi consultation meetings (final model) |
|---|---|---|---|---|
| Child | Outcomes | "Small for gestational age" added and visualization of "low birth weight" changed |  |  |
|  |  | "Short-, mid- and long-term morbidities" changed to "neonatal, infant, <5 and older child/adult morbidities" | "Neonatal, infant, <5 and older child/adult morbidities" changed to "Other morbidities in neonates, infants, <5, older children and adults" | Shape of "Other morbidities in neonates, infants, <5, older children and adults" changed from large arrow to an oval shape as other outcomes |
|  |  | "CHILD with *in utero* exposure to P.falciparum" changed to "CHILD Presence of P.falciparum in utero" | "CHILD Presence of P.falciparum in utero" relabelled to "CHILD Presence of/ exposure to P.falciparum in utero" |  |
|  |  |  | "Neonatal, infant and <5 mortality" combined into 1 large outcome box and relabelled as "Neonatal, infant, <5 mortality & mortality in older children & adults" | green colour added to box "Neonatal, infant, <5 mortality & mortality in older children & adults" |
|  |  |  | "Modified incidence of malaria in <5" changed to "Increased incidence of malaria in <5" |  |
|  |  |  |  | "Neurocognitive development impairment in <5" relabelled to "Neurocognitive &physical development impairment in <5" |
|  | Relationship | Arrow from "neonatal, infant, <5 and older child/adult morbidities" to "neonatal, infant and <5 mortality" added | Starting position of arrow from "neonatal, infant, <5 and older child/adult morbidities" to "neonatal, infant and <5 mortality" changed from back of the box to the middle |  |
|  |  |  |  | Arrows from 1) "CHILD Presence of/exposure to P. falciparum in utero", 2)"Foetal anaemia", 3) "Congenital malaria"to "Other morbidities in neonates, infants, <5, older children and adults"added |
| Other | Design | Symbols in addition of colour code for the timing of health effects added |  |  |
|  |  | Red on timeline changed from "neonate/ 28 days postpartum" to "neonate/mother 28/42 days postpartum" to reflect the postpartum period of 42 days during which maternal deaths are counted | description box moved below model |  |
|  |  | Design of certain arrows and lines changed to help with distinguishing them | Design of arrows and lines changed again as previous change was confusing to expert |  |
|  | Stratifiers | "Transmission intensity" added as a stratifier to the next round in addition to "HIV status","Gravidity" and "Timing of exposure" | Out of the four stratifiers ("Transmission intensity", "HIV status","Gravidity" and "Timing of exposure"), experts voted "Transmission intensity" and "Gravidity" as most important. | Description box listing potential other stratifiers added below the model |

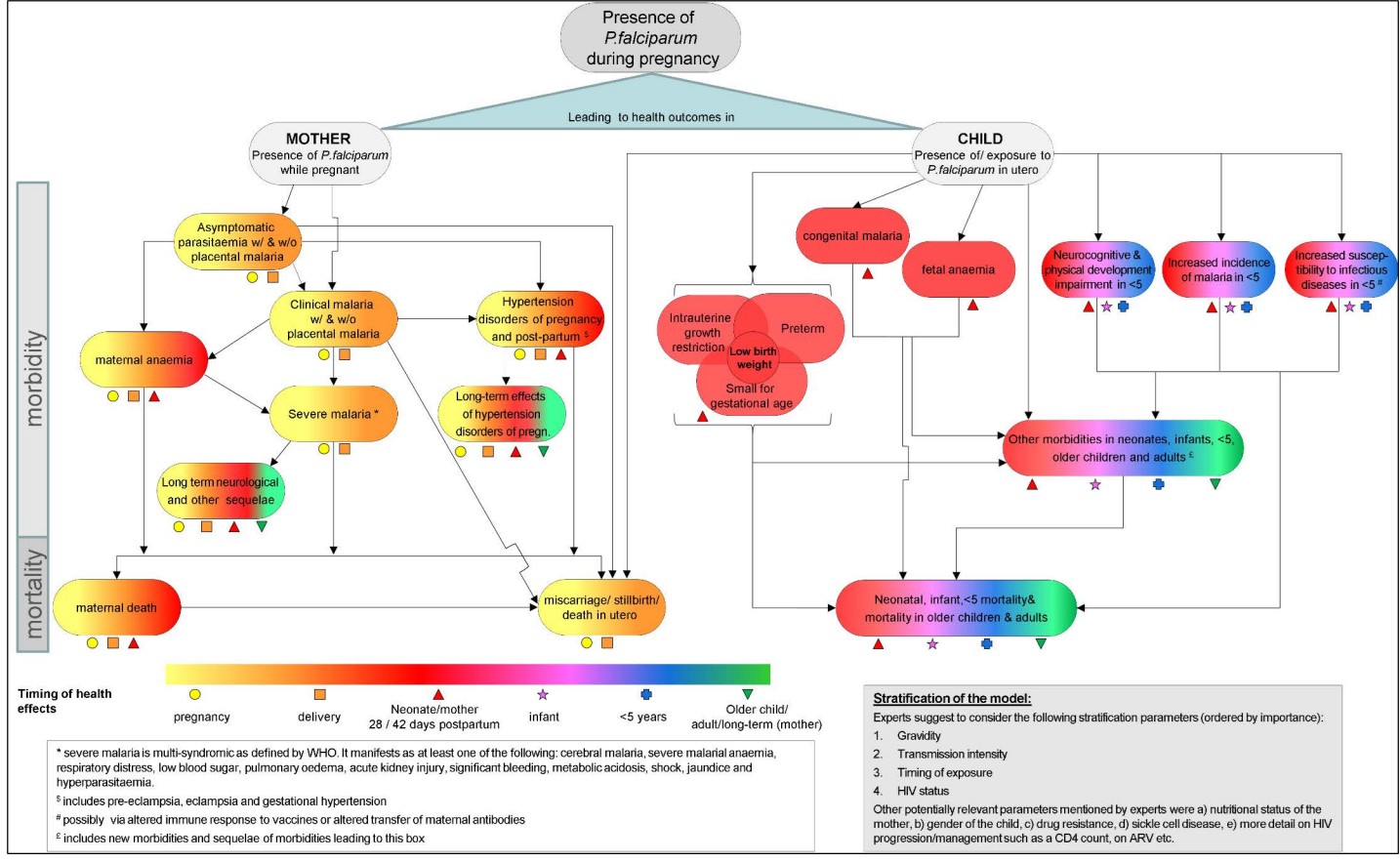

**Fig 4. Final model (after Delphi consultation meetings).** Fig 4 shows the final agreed disease policy model of malaria during pregnancy following two consensus meetings attended by nine experts. The model combines both maternal and child outcomes. Outcomes were divided into maternal and child outcomes. Outcomes affecting morbidity are shown at the top, while mortality outcomes at the bottom. The colour and shape coding of the outcomes represents the different timings of the health effects. Abbreviations: w/ = with; w/o = without; WHO = World Health Organization.

rely heavily on assumptions made in the Global Burden of Disease studies [24]. For example, estimating DALYs arising from "Long-term neurological and other sequelae" could potentially be difficult as long-term follow up data are lacking from malaria trials, requiring assumptions. Likewise, not all manifestations of severe malaria are equally associated with mortality or long-term morbidities, ultimately affecting the DALY calculation. Placental malaria and asymptomatic parasitaemia were included in the model after the first consultation round, because for the experts, in particular epidemiologists, it is important to have these intermediate and often reported outcomes represented in the model for completeness and to depict important pathways.

Experts expressed very differing views regarding the inclusion of hypertension disorders of pregnancy, mostly shaped by different levels of awareness. The votes as well as the comments provided in the Delphi consultation indicated that experts working in lower endemicity settings were more aware of the link between malaria during pregnancy and hypertension disorder during pregnancy. The consensus meetings provided a useful platform to discuss these differences and the supporting evidence. Evidence from both Asia [25] and Africa points to an association, with a meta-analysis including four case-control studies from Africa suggesting that women experiencing malaria during pregnancy had 2.7 times higher odds of developing gestational hypertension disorder compared with those who did not [26]. Ideally, the model should also

differentiate hypertension disorder of pregnancy further into pre-eclampsia and eclampsia, but experts agreed that this level of detail will be difficult to find in the currently available data and should be sought in the future. Experts commented that more effort should be made to collect data on hypertension disorder in pregnancy as blood pressure measurement is often omitted from clinical trial databases, despite being routinely checked. As above for "long-term neurological and other sequalae" the estimation of DALYs arising from hypertension disorder of pregnancy will at this point in time require some assumptions on incidence and disability weights, as hopefully more empirical evidence emerges in the future.

Some of the morbidities and outcomes can have lifelong consequences and be progressive. For example, "neurocognitive and physical development impairment in <5" will impact the child for their entire life and influence their educational achievement and productivity. Likewise, for women with severe malaria who develop severe anaemia and require a blood transfusion, there is a risk of blood supply contamination, which then increases the risk of a wide range of other morbidities associated with other infectious diseases. While it will not be possible to quantify these future consequences for a cost-effectiveness analysis with currently available data, it is certainly important to create awareness of the potential long-term health problems.

This study identified four important stratifiers: gravidity, transmission intensity, timing of exposure and HIV status. However, other variables may be important in certain analyses. Examples could be the sickle cell trait or the gender of the baby. At this point in time insufficient data are available to differentiate the consequences of the timing of exposure (e.g., first versus second, third trimester). HIV status also requires further disaggregation of the data such as the CD4 count or whether the woman is receiving antiretrovirals. Currently, cost-effectiveness models of chemoprevention for malaria during pregnancy will naturally stratify by HIV status as HIV positive and negative women receive different prevention interventions.

Currently, the model does not include potential treatment or prevention interventions to ensure it is widely applicable for different purposes. Depending on the type of intervention study and context, further outcomes might need to be incorporated into the model. Examples are side-effects of any drugs, or effects on other diseases, such as HIV transmission from mother to child in HIV positive women. Studies using QALYs might also need to include patient's health perception, perceived quality of life and future outlook into the model.

Finally, an important outcome of this study is to identify areas where data are scarce and share these with the research community, to raise awareness of the need for comparable outcome measures reported by trials.

A number of these points highlight the urgent need for more granular data, and experts felt that despite the complexity of the model it was useful for creating awareness of the wide range of outcomes that can be averted by preventing pregnant women from being exposed to *P. falciparum*. More detailed data in the future should allow a move away from one size fits all models to models that are more adaptable and fluid.

## Conclusions

This study has demonstrated a more inclusive approach to developing disease policy models capable of assisting in the design of clinical trials (and other policy evaluations) and their associated health economic analysis. In so doing, we believe that this integrated approach should become the gold-standard for disease modelling designed to inform health policy in different countries and contexts. Co-development ensures wider perspectives are incorporated into the model than is usually possible for a single academic team, which should ensure the resulting model is more robust and fit for purpose. A robust modelling co-design approach will also help identify data gaps, ensuring these are not overlooked as the modelling proceeds to the implementation phase.

## Supporting information

**S1 Text. Detailed methods and results.**
(DOCX)

**S2 Text. Questionnaire Delphi consultation round 1.**
(PDF)

**S3 Text. Questionnaire Delphi consultation round 2.**
(PDF)

**S4 Text. Summary report Delphi consultation round 1.**
(PDF)

**S5 Text. Summary report Delphi consultation round 2.**
(PDF)

**S6 Text. Summary of consensus meetings.**
(PDF)

## Acknowledgments

The authors would like to take this opportunity to acknowledge and thank the members of the Delphi Panel and Steering group for giving up their precious time so selflessly to share their invaluable wisdom with the study team.

**Delphi panel members (**in alphabetical order): Prof. Grant Dorsey (University of California San Francisco, United States of America), Prof. Kevin Kain (University of Toronto, UHN-Toronto General Hospital, Canada), Dr. Kassoum Kayentao (Malaria Research and Training Center-University of Sciences, Techniques, and Technologies of Bamako, Mali), Prof. Feiko ter Kuile (Liverpool School of Tropical Medicine, United Kingdom), Prof. Rose McGready (Shoklo Malaria Research Unit, Thailand), Dr. George Mtove (National Institute for Medical Research, Tanzania), Dr. Catherine Pitt (London School of Hygiene and Tropical Medicine, United Kingdom), Prof. Stephen Rogerson (University of Melbourne, Australia), Dr. Makoto Saito (WorldWide Antimalarial Resistance Network), Dr. Steve Taylor (Duke University, United States of America), Prof. Halidou Tinto (Institut de Recherche en Sciences de la Santé, Clinical Research Unit of Nanoro, Burkina Faso) and Dr. Holger Unger (Menzies School of Health Research, Darwin, Australia).

**Steering group members**: Prof. Kara Hanson (London School of Hygiene and Tropical Medicine), Prof. Andy Briggs (London School of Hygiene and Tropical Medicine), Prof. Feiko ter Kuile (Liverpool School of Tropical Medicine), Silke Fernandes (London School of Hygiene and Tropical Medicine)

## Author contributions

**Conceptualization:** Silke Fernandes, Andrew Briggs, Kara Hanson.

**Data curation:** Silke Fernandes.

**Formal analysis:** Silke Fernandes.

**Funding acquisition:** Kara Hanson.

**Investigation:** Silke Fernandes.

**Methodology:** Silke Fernandes, Andrew Briggs, Kara Hanson.

**Project administration:** Silke Fernandes.

**Supervision:** Andrew Briggs, Kara Hanson.

**Validation:** Silke Fernandes, Kara Hanson.

**Visualization:** Silke Fernandes.

**Writing – original draft:** Silke Fernandes, Kara Hanson.

**Writing – review & editing:** Silke Fernandes, Andrew Briggs, Kara Hanson.

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
