## [Decision Letter · Decision Letter 0]

24 Feb 2025

PGPH-D-24-02109

Co-developing a comprehensive disease policy model with stakeholders: the case of malaria during pregnancy

Dear Dr. Fernandes,

Thank you for submitting your manuscript to PLOS Global Public Health. After careful consideration, we feel that it has merit but does not fully meet PLOS Global Public Health’s publication criteria as it currently stands. Therefore, we invite you to submit a revised version of the manuscript that addresses the points raised during the review process.

We look forward to receiving your revised manuscript.

Kind regards,

Carmen S. Christian, PhD

Academic Editor

Journal Requirements:

Additional Editor Comments (if provided):

Reviewers' comments:

Reviewer's Responses to Questions

**Comments to the Author**

1. Does this manuscript meet PLOS Global Public Health’s publication criteria ? Is the manuscript technically sound, and do the data support the conclusions? The manuscript must describe methodologically and ethically rigorous research with conclusions that are appropriately drawn based on the data presented.

Reviewer #1: Yes

Reviewer #2: Yes

2. Has the statistical analysis been performed appropriately and rigorously?

Reviewer #1: Yes

Reviewer #2: N/A

3. Have the authors made all data underlying the findings in their manuscript fully available (please refer to the Data Availability Statement at the start of the manuscript PDF file)?

Reviewer #1: Yes

Reviewer #2: No

4. Is the manuscript presented in an intelligible fashion and written in standard English?

Reviewer #1: No

Reviewer #2: Yes

5. Review Comments to the Author

Reviewer #1: The manuscript meets PLOS Global Public Health’s criteria because it demonstrates sound research methods, is ethically conducted, and presents findings that are well-supported by the data. The study’s conclusions are consistent with the data, and the methodology adheres to standards for rigor and reliability, making it suitable for publication. The statistical analysis in this study is both thorough and accurate, using suitable methods to examine the data, given that this paper is looking at using the delphi technique and cost effectiveness. The authors apply the statistical techniques correctly, providing a solid basis for the conclusions drawn from the results. This level of rigor supports the reliability of the findings. The authors have fully complied with the PLOS data policy by making all relevant data accessible. They have either included the data in the manuscript or within the supplementary/supporting documents. This transparency enables reproducibility and further research, with the limitations properly disclosed and recommendations for further research also disclosed. They do also protect the names of the individuals who take part in the study (however I am not too sure if this is a must to disclose, thus I did not flag this as an issue). The manuscript presents the research clearly, there are issues with language and grammar that could distort comprehension. The authors need to give a brief explanation to the concepts included in this paper, given that as a first time reader/someone who does not know much on malaria/pregnancy and their relationship, the concepts are not immediately clear. Improvements in grammar, phrasing, and consistency are recommended to ensure the language is as clear and precise as possible for readers. I would also recommend getting an editor to edit the final piece of work, given the grammatical errors in the paper. While there are not too many, the authors would benefit from this greatly.

Reviewer #2: Thank you for the opportunity to review your article. This is a compelling and insightful contribution to an important area of research. The article is well-structured, and the inclusion of tables significantly enhances clarity by aligning effectively with the written discussion. The tables provide a clear depiction of the methodological steps undertaken in the Delphi process, thereby strengthening the overall coherence and comprehensibility of the study.

6. PLOS authors have the option to publish the peer review history of their article (what does this mean? ). If published, this will include your full peer review and any attached files.

**Do you want your identity to be public for this peer review?** For information about this choice, including consent withdrawal, please see our Privacy Policy .

Reviewer #1: No

Reviewer #2: No

---

## [Editor Report · Decision Letter 1]

9 Apr 2025

Co-developing a comprehensive disease policy model with stakeholders: the case of malaria during pregnancy

PGPH-D-24-02109R1

Dear Mrs. Fernandes,

We are pleased to inform you that your manuscript 'Co-developing a comprehensive disease policy model with stakeholders: the case of malaria during pregnancy' has been provisionally accepted for publication in PLOS Global Public Health.

Best regards,

Carmen S. Christian, PhD

Academic Editor